# Dietary Interventions Ameliorate Infectious Colitis by Restoring the Microbiome and Promoting Stem Cell Proliferation in Mice

**DOI:** 10.3390/ijms23010339

**Published:** 2021-12-29

**Authors:** Ishfaq Ahmed, Kafayat Yusuf, Badal C. Roy, Jason Stubbs, Shrikant Anant, Thomas M. Attard, Venkatesh Sampath, Shahid Umar

**Affiliations:** 1Department of Surgery, University of Kansas Medical Center, Kansas City, KS 66160, USA; iahmed@kckcc.edu (I.A.); kyusuf@kumc.edu (K.Y.); broy@kumc.edu (B.C.R.); 2Department of Internal Medicine, University of Kansas Medical Center, Kansas City, KS 66160, USA; jstubbs@kumc.edu; 3Cancer Biology Department, University of Kansas Medical Center, Kansas City, KS 66160, USA; sanant@kumc.edu; 4Department of Pediatrics and Gastroenterology, Children’s Mercy Hospital, Kansas City, KS 66160, USA; tmattard@cmh.edu (T.M.A.); vsampath@cmh.edu (V.S.)

**Keywords:** intestinal stem cells, lgr5, gut microbiome, *Citrobacter rodentium*, colitis, Pectin, Tributyrin, short-chain-fatty-acids (SCFAs), bacterial infection, dietary intervention

## Abstract

Decreases in short-chain-fatty-acids (SCFAs) are linked to inflammatory bowel disease (IBD). Yet, the mechanisms through which SCFAs promote wound healing, orchestrated by intestinal stem cells, are poorly understood. We discovered that, in mice with *Citrobacter rodentium* (CR)-induced infectious colitis, treatment with Pectin and Tributyrin diets reduced the severity of colitis by restoring *Firmicutes* and *Bacteroidetes* and by increasing mucus production. RNA-seq in young adult mouse colon (YAMC) cells identified higher expression of Lgr4, Lgr6, DCLK1, Muc2, and SIGGIR after Butyrate treatment. Lineage tracing in CR-infected *Lgr5-EGFP-IRES-CreERT2/ROSA26-LacZ* (*Lgr5-R*) mice also revealed an expansion of *LacZ*-labeled Lgr5(+) stem cells in the colons of both Pectin and Tributyrin-treated mice compared to control. Interestingly, gut microbiota was required for Pectin but not Tributyrin-induced Lgr5(+) stem cell expansion. YAMC cells treated with sodium butyrate exhibited increased Lgr5 promoter reporter activity due to direct Butyrate binding with Lgr5 at −4.0 Kcal/mol, leading to thermal stabilization. Upon ChIP-seq, H3K4me3 increased near Lgr5 transcription start site that contained the consensus binding motif for a transcriptional activator of Lgr5 (SPIB). Thus, a multitude of effects on gut microbiome, differential gene expression, and/or expansion of Lgr5(+) stem cells seem to underlie amelioration of colitis following dietary intervention.

## 1. Introduction

The gastrointestinal tract is home to trillions of microbes called microbiota [1] that protect against enteropathogens, contribute to normal immune function, and maintain homeostasis in the gut [1]. Maintaining this functional balance is crucial for sustaining normal intestinal mucosa physiology, as disruptions have been linked to the pathophysiology of various gastrointestinal complications, including inflammatory bowel disease (IBD) [2].

The gut microbiota is known to secrete small molecules and metabolites such as short-chain-fatty-acids (SCFAs), which play an essential role in maintaining intestinal functions and homeostasis [3,4,5]. SCFAs, including acetate, Butyrate, and propionate, are produced by the fermentation of fiber diet, e.g., Pectin, in the colon [3,4,5]. Indigestible carbohydrates in the diet serve as fermentation substrates for anaerobic gut bacteria, which produce SCFAs and other gases as end products [3]. It is known that IBD patients have lower fecal concentrations of SCFAs compared with healthy control subjects and are often linked to higher disease severity [6]. Diet and ingested microbes can potentially change the gut environment by creating competition amongst gut microbiota for resources, leading to the alteration of the microbial community. Many health conditions, including IBD, have been associated with specific alterations in gut microbiota composition, which affects host energy, metabolism, and immunity [1,7,8,9,10,11,12].

The gut’s protective effects against external insults depend on the activity of a functional intestinal epithelium which is constantly maintained by the continuous and rapid proliferation of undifferentiated cells located within the crypts [13]. The intestinal epithelium is renewed every 3 to 5 days, facilitated by tissue-specific stem cell markers located at the base of the crypts [14]. Leucine-rich repeat-containing G protein-coupled receptor 5 (LGR5) is a protein that has been discovered as a marker of crypt basal columnar stem cells in the intestine [14]. Studies have also highlighted that LGR5 is closely related to two additional receptors, leucine-rich repeat-containing G protein-coupled receptor 4 (LGR4) and leucine-rich repeat-containing G protein-coupled receptor 6 (LGR6), which share around a 50% identity and represent a structurally distinct category of transmembrane receptors [15,16] with specific functions in embryonic development and adult tissue homeostasis [15,16].

In many adult tissues such as the intestinal epithelium with rapid proliferation, renewal, and regeneration rates, Wnt signaling is well documented as a master of epithelial renewal, as seen by the rapid fall in epithelial regeneration following reduced Wnt activity [17]. Studies have also shown that Wnt signaling is the main driver of cell proliferation in the gut [18,19,20]. Likewise, a study revealed that LGR4 and LGR5 potentiate Wnt/β-Catenin signaling in response to R-Spondin proteins [15].

*Citrobacter rodentium* (CR) is a Gram-negative enteric bacterium that causes Transmissible Murine Colonic Hyperplasia (TMCH) in mice characterized by colon hyperplasia, a loss of goblet cells, and increased inflammation depending upon the genetic background [21,22]. CR is a member of attaching and effacing (A/E) family of bacterial pathogens and models human *enteropathogenic Escherichia coli* (EPEC), and *enterohaemorrhagic Escherichia coli* (EHEC) and can regulate the composition of commensal microbiota, the integrity of epithelial barrier, mucosal healing, and inflammation, making it a robust model to study the pathogenesis of human intestinal disorders including Crohn’s disease, ulcerative colitis (UC), and dysbiosis [23,24,25].

Previous findings have revealed the role of dietary fibers such as Pectin in maintaining gastrointestinal homeostasis [26] by releasing metabolites such as Butyrate and SCFAs [3,26]. However, there is little understanding of the role of dietary fibers in regulating the expression of stem cell markers in mitigating colitis. Thus, this study aimed to determine the effect of Pectin and Tributyrin diets on alleviating the effects of CR-induced colitis by regulating crypt stem cell markers Lgr4/5/6 to regenerate epithelial crypts, modifying the microbiome to restore homeostasis alongside augmenting tissue repair in infected mice.

## 2. Results

### 2.1. Pectin and Tributyrin Diets Ameliorate Symptoms of Colitis 

Mice infected with CR developed colitis characterized by colon shortening, weight loss, and general inflammation of the colonic tissue (Figure 1Ai,Aii). To characterize the integrity of the colonic epithelial layer, we quantified the permeability of orally administered FITC-dextran in serum. We found increased levels of serum FITC-dextran in CR-infected mice (Figure 1Aiii). Pectin (CR+Pec) and Tributyrin (CR+Tbt) diets (diets hereafter), however, showed a significant decrease in serum FITC-dextran (Figure 1Aiii). To further characterize the protective effect of diets, we performed electron microscopy of mucosal epithelium. We observed loss of microvilli and increased intercellular distance in CR compared with N. Contrarily, in diet samples, microvilli were restored to some extent and intercellular distances were minimized (Figure 1Bi,Bii). H&E staining revealed longer crypts and immune cell infiltration in CR-infected mice compared to healthy control (N). However, the mice infected with CR and fed with diets showed a decreased infiltration of immune cells (Figure 1Ci,Cii). We next analyzed the effect of diets on cell proliferation during CR infection by staining colonic sections for the proliferation marker Ki67. Our results showed an increased proliferation of crypt cells in CR-infected mice compared to the N. Diets maintained the proliferation state of colon crypts (Figure 1Di,Dii). These results indicate that CR infection-mediated infectious colitis is characterized by colon inflammation and shortening, weight loss in the animals, increased FITC, increased crypt proliferation, and immune cell infiltration. The Pectin and Tributyrin diet, on the other hand, reduced disease severity by restoring colon characteristics, maintaining mucosal epithelium, and preserving crypt proliferation.

### 2.2. Pectin and Tributyrin Diets Enhanced MUC2 Secretion in CR-Infected Groups

Goblet cells are abundant in the colon than in other areas of the gastrointestinal tract and are important for protecting the integrity of the intestinal mucosal barrier against pathogens. Likewise, goblet cells are also responsible for secreting intestinal antimicrobial peptides and mucins. We performed PAS/Alcian blue staining on the Carnoy’s fixed colonic sections to identify all intracellular mucin glycoproteins [27,28]. As expected, there was a loss of goblet cells and the inner mucus layer thickness in CR-infected mice compared with N, while CR-infected mice put on diets were able to restore them (Figure 2A,B). Appendix A represents the average number of goblet cells/crypt. 

Mucin-2 (Muc-2) is a major colonic mucin secreted by goblet cells and is protective against CR infection [29]. We utilized the immunofluorescence of colon sections using the anti-Muc2 antibody to ascertain the expression of Muc-2. We observed a significant loss of Muc-2 secretion in CR-infected mice compared with controls (Figure 2C,D). Diets, however, enhanced the secretion of Muc2 significantly (Figure 2C,D). Appendix A represents average number of Muc2+ cells/crypt. Muc-2 depletion during infection is also correlated with an impaired intestinal mucus layer [30]. To investigate whether the impaired mucus layer caused by the infection increased bacterial access to epithelium and subsequently contributed to colitis, we performed staining to detect luminal bacteria via fluorescence-in-situ-hybridization (FISH) with the universal bacterial probe EUB338. Staining of Carnoy--fixed colon sections indicated that the bacteria in the CR-infected samples resided on the epithelial surface, which was in contrast with N, where the mucus layer separated bacteria from the epithelial layer. The addition of the diet, however, increased the distance between the bacteria and epithelial barrier due to mucus production and hence disallowing bacteria to bind with the epithelial surface (Figure 2E,F). Together, these results suggest that Pectin and Tributyrin diets alleviate colitis-associated goblet cell depletion by increased secretion of Muc2.

### 2.3. Microbial Diversity and Composition Changes Following Pectin Diet 

CR infection has been associated with dysbiosis of colonic microbiota [31]; likewise, fiber diets have been shown to release SCFAs through bacterial fermentation, which promotes the growth of beneficial bacteria in the gut [32]. To investigate the impact of diets on gut microbiota composition and whether they can reverse the CR-induced dysbiosis, we analyzed colon microbiota dynamics of mice fed with Pectin compared with the control diet using bacteria16S rRNA gene sequencing. We observed a significant expansion of bacteria in the phylum *Proteobacteria* coupled with a concomitant decrease in the phylum *Bacteroidetes* and *Firmicutes* in the CR-infected mice (Figure 3A). CR+Pec mice were highly responsive to diet change and decreased the growth of *Proteobacteria* while increasing the abundance of *Bacteroidetes* and *Firmicutes* in a pattern similar to the control groups (Figure 3A). To better understand the impact of Pectin on the microbial community of CR-infected mice, we studied the correlation between Pectin diet and the taxonomic composition at the family and operational taxonomic unit (OTU) level. We observed an overgrowth of *Enterobacteriaceae* and *Bacteroidaceae* alongside a decrease in the S24-7 family of bacteria in CR-infected mice compared with control (Figure 3B). In contrast, CR+Pec mice showed a significant diminution of *Enterobacteriaceae,* an increase in S24-7 bacteria, as well as a corresponding increase in the abundance of Butyrate-producing bacteria such as *Lachnospiraceae*, *Lactobacillaceae,* and *Ruminococaceae* (Figure 3B). Alpha-diversity, which describes diversity within a sample and is measured by the PD Whole Tree (Figure 3C), observed OTUs (Figure 3D), and the Chao1 index (Figure 3E), all decreased significantly with CR infection. However, the Pectin diet increased these indices significantly (*p* < 0.05) (FDR ≤ 0.05) (Figure 3C–E). β-diversity describes the microbiome diversity between samples and can be described using principal component analysis (PCoA) based on the weighted and unweighted UniFrac distance (Figure 3F), LEfSe or linear discriminant analysis (LDA) using a horizontal bar graph (Figure 3G), or by using a LEfSe cladogram (Figure 3H). The distinct clustering of the N group, visible through the β-diversity analysis, indicated abundant bacterial species in the N group that significantly diminished with CR infection. Pectin-treated samples clustered better than CR, indicating increasing bacterial abundance (Figure 3F–H). The PCoA showed differences in the cluster of bacteria between the three different groups; moreover, in the second plot (Figure 3F), which was from a different animal, there is a similarity in the cluster observed in the control and Pectin-fed group suggesting that the Pectin diet may have restored the bacteria groups to normal (Figure 3F). Taxonomic analysis also showed that increased diversity of bacteria in the N group diminished with CR infection (Figure 3G–H). These results suggest that the Pectin diet may have dual role in either preventing the loss of key bacteria or restoring microbial diversity in the infected mice.

Investigation with the Tributyrin diet produced almost similar results with a decrease in relative abundance of *Firmicutes* but an increase in *Proteobacteria* in CR-infected mice compared to control (Appendix A). In response to Tributyrin, while *Bacteroidetes* were not affected, we did see a relative increase in *Firmicutes* with a concomitant reduction in *Proteobacteria* (Appendix A). Lack of detailed alpha- and β-diversity measurements in the Tributyrin-fed group remains a limitation of this study.

### 2.4. Pectin Fermentation Product Butyrate Induces Differential Expression of Genes Involved in Epithelial Regeneration and Repair

In order to have a better understanding of how diet mitigates infectious colitis together with delineating the effect of SCFAs such as Butyrate on epithelial wound healing and repair, we decided to conduct an RNA-seq analysis on young adult mouse colon cells (YAMCs) treated with sodium butyrate in vitro. We present volcano plots and heatmaps of differentially expressed genes that were upregulated and statistically different in the three groups (Table 1). We noticed that a larger pool of genes was differentially expressed in the CR+Butyrate (CR+B) vs. control group and CR+B vs. CR group (Appendix A). Thousands of differentially expressed genes were identified but were streamlined to specific genes involved in cytokine production, transcription regulation, protective effect, and epithelial repair, with *p*-value (*p* ≤ 0.05), false discovery rate FDR ≤ 0.05, and a fold change greater than 1.5. (Appendix A). We found upregulation of anti-inflammatory genes IL18, IL4, and IL33 after Butyrate treatment (Butyrate vs. CR). These genes had decreased expression in the CR vs. control group (Figure 4). Likewise, we observed the downregulation of the proinflammatory gene IL6 in the Butyrate vs. CR group, which, in contrast, was upregulated in the CR vs. control (Figure 4). We also identified the upregulation of tissue-specific stem cell markers LGR4 and LGR6 in both the Butyrate vs. control group as well as Butyrate vs. CR group (Figure 4, Appendix A). Contrarily, the expression of these stem cell markers declined in the CR-infected group (Figure 4). Wnt ligands WNT2B, WNT7A, and receptor FZD9 also had increased expression after Butyrate treatment. In the same light, Wnt pathway agonists RSPO1 and RSPO3 and co-receptors LRP5 and LRP6 had increased expression in the Butyrate vs. CR group (Figure 4), suggesting a possible upregulation of the Wnt pathway after Butyrate treatment.

DCLK1, or doublecortin-like kinase 1, is a marker of intestinal tuft cells that has been shown to improve epithelial repair responses, limiting bacterial invasion into the mucosa [33]. We found upregulation of DCLK1 in the Butyrate vs. CR group and a decreased expression of DCLK1 after CR infection (Figure 4 and Appendix A). MUC-2, which has been primarily associated with epithelial repair and mucosal barrier sustenance [29,30], showed increased expression after Butyrate treatment (Figure 4 and Appendix A). NOTCH3 and NOTCH4 receptors alongside JAG1 ligand of the Notch pathway were upregulated in the Butyrate vs. CR group (Figure 4 and Appendix A) and contrarily downregulated in the CR vs. control group. Wnt and Notch signaling facilitate wound healing in the cutaneous layer of the skin [34]; in lieu of this, we opine that upregulation of Wnt and Notch pathways, increased expression of DCLK1, MUC-2, nucleotide-binding oligomerization domain 2 (NOD2), and anti-inflammatory markers all promote epithelial regeneration after CR infection.

### 2.5. Pectin and Tributyrin Diets Regulate Lgr5 Expression

Our RNA-seq data revealed the upregulation of LGR4 and LGR6 but not LGR5 after Butyrate treatment. However, previous studies have confirmed that LGR5 is highly related to LGR4 and LGR6, with over 50% similarity in structure [14,15]; they are also grouped together as type B receptor family of proteins or the G-protein-coupled, 7-transmembrane (7TM) family of proteins [14,15,16]. In view of this, we decided to test whether the beneficial effect of diets on colonic epithelium is related to the increased expression of Lgr5+ stem cells. We examined the effects of diets on Lgr5+ colonic stem cells in *Lgr5CreERT2/Rosa26LacZ* reporter (*Lgr5-R*) mice [14] during CR infection (Figure 5A). The administration of Pectin and Tributyrin diets significantly enhanced LacZ staining in the colons of these mice when compared with CR, indicating that Pectin and Tributyrin diets could induce the expression of Lgr5 in the colonic crypts (Figure 5B). Appendix A represents average number of *LacZ*+ cells/crypt. Additionally, quantitative PCR of Lgr5 in whole animal tissues revealed a decline in Lgr5 expression after CR infection with a slight restoration following diet intervention (data not shown). We next verified whether the induction of Lgr5 by diets required the presence of microbes. To do so, we gave a cocktail of antibiotics to Lgr5-R mice infected with CR and fed them with a 6% Pectin (CR+Pec) or Tributyrin (CR+Tb) diet, respectively. The antibiotics treatment significantly reduced *LacZ* staining in the crypts of CR+Pec but not in CR+Tb mice, indicating that Lgr5 induction was indeed driven by microbiota (Figure 5C). Appendix A represents the average number of *LacZ*+ cells/crypt. To see whether antibiotic treatment was able to slow down the crypt proliferation, we performed Ki-67 staining on the colon sections. As expected, CR-infected mice fed with a Pectin diet and treated with antibiotics showed a decrease in Ki-67 staining compared with control (Figure 5D), while antibiotics did not have a similar effect on the mice fed on the Tributyrin diet (Figure 5D). Appendix A represents the average number of Ki-67+ cells/crypt. We acknowledge that the lack of 16S rRNA gene sequencing in antibiotics-treated samples is a limitation of the study.

### 2.6. Butyrate Increases LGR5 Promoter Activity

To see the effect of SCFA Butyrate directly on Lgr5 binding, we performed an Lgr5 promoter activity assay. Sodium butyrate increased Lgr5 promoter reporter activity dose-dependently (1–5 mM) then plateaued at 10 mM concentration (Figure 6A). To explore the ability of Butyrate to bind to and confer stability to Lgr5, we performed molecular docking analysis and cellular-thermal-shift-assay (CETSA), respectively. We used the Autodock Vina software program to perform molecular docking to analyze the binding of Butyrate in the protein cavity of human Lgr5. The X-ray crystal structure of Lgr5 (PDB ID: 4UFR) was used for docking. Our docking data revealed that Butyrate binds within the protein cavity of Lgr5 (Binding energy = −4.0 Kcal/mol) and forms hydrogen bonds with Asp290 (2.3 Å) and Phe288 (2.0 & 2.6 Å) (Figure 6B,C). This interaction of Butyrate with Lgr5 at 2 μM-1mM led to thermal stabilization of the Lgr5 protein at 40–60 °C in a cellular thermal shift assay (Figure 6D,E).

### 2.7. Butyrate Promotes Active Transcription of Lgr5 through SPIB

Histone modifications are well-recognized mechanisms that mediate the alteration of gene expression [35]. We performed ChIP-sequencing using young adult mouse colon cells (YAMCs) with antibodies specific to histone H3-lysine 27 acetylation (H3K27Ac), histone H3-lysine 9 acetylation (H3K9Ac), and histone H3-lysine 4 methylation (H3K4me3) modification. The number of significantly enriched regions detected for each histone modification marker is described in Figure 7E. The distribution of differential binding events of the H3K27ac marker compared across N, CR, and CR-B is given in Appendix A. CR constituted the highest number of unique H3K27ac sites (50.87%), followed by CR-B (37.12%) and N (6.4%) (Appendix A). There were several sites enriched for the H3K27Ac marker around a 100 kb region from the transcription start site (TSS) of the Lgr5 gene in both CR (10 sites) and CR-B (8 sites) but not in N (2 sites) (Figure 7A). Two of these sites upstream (12,697 bp and 79,049 bp from the TSS) in CR group and one site overlapping (5972 bp from the TSS) in the CR-B group contained consensus binding motif for Neurogenin 1 (NEUROG1), which is known to regulate Lgr5 [36]. We also found that histone H3 tri-methylation at lysine 4, a mark of active transcription [35], was increased near Lgr5 TSS in CR+B-treated cells (Figure 7B). Several sites were enriched for the H3K4me3 marker around a 100 kb region from the TSS of the Lgr5 gene in all three conditions (N: 7, CR: 8, CR-B: 9) (Figure 7B). Two of these sites were upstream (8357 bp and 46713 bp from the TSS) of Lgr5 in N and four sites (one 25831 bp upstream and three downstream overlapping the gene at 26813 bp, 38716 bp, 69891 bp from the TSS) in CR+B contained the consensus binding motif for the Spi-B transcription factor (SPIB), a transcriptional activator which is known to induce Lgr5 expression [37,38]. The differential expression of SPIB was also seen in the RNA-seq data, wherein decreased expression was observed in CR vs. control group, which was upregulated after Butyrate treatment (Appendix A). This suggests that Butyrate may promote active transcription of Lgr5 by inducing the expression of SPIB.

The distribution of differential binding events for H3K9Ac, a marker of active transcription [35], was also compared across N, CR, and CR-B and is given in Appendix A. CR contained the highest number of unique H3K9ac sites (92.2%), followed by CR-B (3.9%) and N (3.0%) (Figure 7C). The Lgr5 gene had two sites upstream (8413 bp and 88142 bp) of its TSS enriched for H3K9ac in CR and contained conserved binding motifs for 14 known TFs (Figure 7D). The site, 88,142 bp upstream from its TSS, contained conserved binding sites for eight TFs, including Neurogenin1 (NEUROG1), a transcriptional regulator of Lgr5 [36].

## 3. Discussion

In this study, we demonstrate that Pectin and Tributyrin diets mitigate infectious colitis through a multitude of events, including recruitment of the stem cell markers Lgr4, Lgr5, and Lgr6 to promote crypt regeneration, partial reversal of bacterial dysbiosis, increased expression of MUC2, DCLK1, NOD2, SIGIRR (single immunoglobulin IL-1R-related molecule), and several other anti-inflammatory genes.

Ulcerative colitis and other intestinal degenerative diseases are common human disorders, as new UC cases seem to be increasing globally [39]; yet, the clinical interventions are limited, as the pathological mechanisms are not very clear. *Citrobacter rodentium* (CR) infection is a robust model to study the pathogenesis of human intestinal disorders including colitis; CR infection leads to weight loss, shortening of the colon, loss of epithelial barrier integrity, bacterial translocation into lamina propria, crypt hyperplasia, goblet cell depletion, loss of inner mucus layer, alteration in microbiota composition and colon inflammation [22,23,26,40,41,42]. Studies have also shown that mice lacking the inflammasome components Nlrp3, Nlrc4, and caspase-1 are hypersusceptible to *Citrobacter rodentium*-induced gastrointestinal inflammation [43].

Diet is the main contributor to the gut microbial composition, diversity, and richness [44,45,46,47,48], and changes in the diet can account for 57% of the variations in microbiota, while genetic variations in the host can account for only 12% [49]. Several studies have shown a strong connection between diet and microbiota, signifying how the composition of different diets directly impacts gut microbiota [45,47,48,50]. In the current study, we observed the predominance of *Enterobacteriaceae* during CR infection. However, the Pectin diet inhibited the growth of Enterobacteriaceae and enhanced the growth of Butyrate-producing bacteria such as *Lachnospiraceae*, *Lactobacillaceae,* and *Ruminococaceae*. This data suggests that the Pectin diet modulates the gut microbiota to resist CR colonization and CR-induced dysbiosis. At the same time, the Pectin diet seems to have either boosted the growth and/or retention of healthy bacteria, particularly those involved in Butyrate production. This phenomenon seems to be duplicated by the Tributyrin diet, although the mechanisms detailing whether Tributyrin’s effect is mediated by Butyrate remain a limitation of this study. We also observed that all diets were able to produce a protective mucus layer on top of the epithelial cells to act as a barrier to pathogens [23]. Increased pathogen susceptibility to CR due to the loss of mucus production in our study parallels previous studies [29], while IBD patients tend to harbor higher levels of mucolytic bacteria that can degrade mucin [51]. This could be attributed to the low intake of dietary fibers with associated decreases in *Firmicutes* and/or *Bacteroidetes* and an overabundance of *Proteobacteria*, all of which are critical in the pathogenesis of IBD [52].

Employing RNA-seq, we present a model of differentially expressed genes after Butyrate treatment to propose a possible mechanism of Butyrate action on mitigating colitis (Figure 4). With the increased expression of MUC-2 after diet (Figure 2 and Figure 4), Butyrate seems to have stimulated MUC-2 secretion to restore the barrier function against bacterial infection in the gut. This is supported by a previous investigation that revealed that MUC-2-deficient mice are vulnerable to colitis and death caused by CR infection [29]. Interestingly, RNA-seq analysis also revealed that genes that exert protective effects in the intestinal epithelium, such as *DCLK1* and *NOD2*, were all upregulated (Figure 4) after Butyrate treatment. DCLK1 is a microtubule-associated protein that marks long-lived, quiescent epithelial tuft cells in the intestine [53], and recent studies have revealed that deletion of DCLK1 led to worsened colitis [33,53]. Likewise, NOD2 modulates innate responses to intestinal microflora by downregulating numerous TLR responses, and the lack of this regulation increases susceptibility to colitis [54].

Cytokines, which are essential mediators of both proinflammatory and anti-inflammatory processes, are crucial in progressing immunological alterations and chronic inflammation in IBD [55]. This study discovered the upregulation of anti-inflammatory cytokines IL33, IL18, and IL4 and the downregulation of proinflammatory cytokine IL6 (Figure 4). IL33 has been described as a dual function protein, however a recent study indicated that IL-33 attenuates chronic intestinal inflammation by promoting epithelial regeneration and weight recovery in infected mice [56]. In the same light, IL18 has been implicated as a dual role cytokine in colitis, and it was proposed that the anti-inflammatory effects of IL-18 may be observed in the early stages of colon inflammation [57]. A study on human IL4-treated regulatory macrophages also revealed that IL4 treatment promoted epithelial wound healing and reduced the severity of colitis in mouse models [58]. Contrarily, IL6 is a proinflammatory cytokine with higher severity of colitis being associated with increased IL6 expression [59,60,61]. A reduction in IL6 expression was observed in the Butyrate vs. control and Butyrate vs. CR-infected group after RNA seq analysis (Figure 4). We speculate that Butyrate treatment promotes epithelial repair through the upregulation of anti-inflammatory genes and the downregulation of proinflammatory markers during CR infection.

Another possible mechanism of Pectin and Tributyrin diets ameliorating colitis may be through the activation of the Wnt pathway. Wnt/β-catenin signaling has been described as the fundamental organizer of epithelial stem cell identity and maintenance [18,62]. Active Wnt signaling is required for epithelial homeostasis in the intestine, and pathway blockage has been associated with crypt loss and tissue degeneration [63]. R-Spondins (RSPO1-4) are Wnt agonists and have been described as the endogenous ligands of the Lgr-expressing (Lgr4/Lgr5/Lgr6) stem cells, with studies demonstrating the critical involvement of Lgr proteins in stem cell homeostasis in the gastrointestinal tract [13,14,15,63]. Our study revealed an increased expression of Wnt receptor and ligands, as well as R-Spondins and Lgr-proteins Lgr4/Lgr5/Lgr6 (Figure 4 and Figure 5), suggesting the upregulation of the Wnt pathway in a bid to repair the damaged epithelium. Furthermore, Lgr5+ stem cells play a significant role in the homeostasis and regeneration of the gut epithelium [13,64,65]. Similarly, lactate plays a vital role in maintaining the stemness of Lgr5+ ISCs by activating the Wnt-β-catenin pathway [66]. Indeed, we found a significant increase in the Lgr5+ stem cell population in the colons of mice fed with Pectin and Tributyrin diets compared to controls. We also found an increase in the active transcription of Lgr5 via SPiB from our ChIP seq analysis that correlates with the previous finding that RankL-induced expression of SpiB is essential for Lgr5 stem cell-derived epithelial precursors to develop into M cells [37]. The increased Lgr5-expressing cells likely contribute to the proliferation and restitution of the inflamed colon. Remarkably, a recent study also revealed that NOD2 mediates LGR5+ intestinal stem cell protection against ROS cytotoxicity [67], suggesting that an increased expression of NOD2, as observed in our study, may also potentiate the activity of Lgr5+ stem cells.

Since SCFAs, including Butyrate, are produced from Pectin through the actions of gut microbiota, and because Butyrate release from Tributyrin requires host lipases [68], we explored the question of whether microbiota regulated stem cells through Butyrate. Our findings indicate that, while both Pectin and Tributyrin diets increased Lgr5+ stem cells, treatment of Pectin-fed mice with antibiotics severely impacted Lgr5 expression, and there was no significant difference in mice fed with the Tributyrin diet and treated with antibiotics. Pectin-fed mice receiving antibiotics also showed significantly less proliferating crypts, and this was not evident in the Tributyrin+Abx group. These results infer that Pectin and Tributyrin diets increased Lgr5 expression in the colonic tissue, leading to the increased proliferation of colonic crypts. We acknowledge that Kaiko et al.’s elegant study recently revealed that the differentiated colonocytes that consumed Butyrate may have prevented it from reaching the proliferating epithelial stem/progenitor cells within the crypt during normal homeostasis [69]. However, we and others have shown that CR infection leads to the remodeling of the mucosa, wherein mitotically active colonocytes could be seen even at the crypt surface [41,70,71]. These cells may continuously be exposed to Butyrate levels that may be sufficient to sustain crypt proliferation through stem cell expansion. This study proposes that a Pectin diet, probably through the generation of SCFAs, protects the colonic epithelium in colitis-infected mice by reducing hyperplasia and restoring gut homeostasis via lgr5+ stem cell expansion. We propose that the crypt is damaged during colon inflammation, resulting in the loss of differentiated cells. In this scenario, we believe that crypt base columnar (CBC) cells use Butyrate to gradually restore homeostasis, as seen in the restoration of Wnt/Notch pathways, which eventually expands Lgr5+ stem cells to promote the renewal of the damaged Crypt.

The signals received from the mesenchymal cells are critical for the maintenance of the stem cell niche. Epithelial response to the disruption of these signals is a critical feature of the intestine’s ability to respond to injury by regulating barrier function, the composition of the microbiota, and mucosal immune homeostasis. We provide evidence that a Pectin diet restores barrier function and mitigates colitis by retaining/restoring healthy bacteria and expanding the stem cell population in the crypt. We also demonstrate for the first time the effect of Butyrate on Lgr5 binding and stabilization via the transcription factor SPIB. With this new discovery, we hypothesize that the supplementation of Butyrate and probably other SCFAs in diets will be highly beneficial in reducing intestinal injury in response to enteric pathogens.

## 4. Methods

### 4.1. Animals

C57Bl/6 and C3H/HeNHsd (C3H) inbred mice were procured from Harlan Laboratories Inc., Indianapolis, IN, USA. C3H mice respond very aggressively to CR infection and are used as an excellent model of infectious colitis [70]. *Lgr5-EGFP-IRES-CreERT2* and *Rosa26LacZ* mice were purchased from Jackson Laboratories. All the mice were maintained in a specific pathogen-free (including Helicobacter and parvovirus) environment and are generally used between 5 and 6 weeks of age. For control groups, either littermates or WT mice of identical backgrounds were used. Animals were placed on diets on the third day after CR infection. Mice were placed into control group (N), CR-infected group (CR), CR-infected and cellulose-fed group (Cell), group fed with Pectin (CR+Pec), and the group fed with the Tributyrin diet (CR+Tbt). A minimum of five animals were used in each group. This study was carried out in strict accordance with the recommendations in the Guide for the Care and Use of Laboratory Animals of the National Institutes of Health. All protocols were approved on 10/20/2017 by the University of Kansas Medical Center Animal Care and Use Committee (IACUC) under the protocol #2017-2425. All mice were maintained in a pathogen-free environment and housed in cages in groups of five animals per cage with constant temperature and humidity and a 12hr/12hr light/dark cycle. All animals always had access to diet and water.

### 4.2. Lineage Tracing

*Lgr5-EGFP-IRES-CreERT2* and *Rosa26LacZ* mice were purchased from Jackson laboratories and cross bred to generate *Lgr5-EGFP-IRES-CreERT2/Rosa26LacZ* (Lgr5.R) mice. Six-to-ten-week-old male and female mice were used for all studies. Cre recombinase was activated in Lgr5.R mice by intraperitoneal injections of tamoxifen (100 μL of 10 mg/mL tamoxifen dissolved in sunflower oil) given on days 6, 7, and 8 post-infection. Dissected colons were fixed with paraformaldehyde, and the *LacZ*+ve progeny of the Lgr5+ve stem cells were visualized in the colon by β-galactosidase staining [15].

### 4.3. Treatments and Humane Endpoints

Transmissible Murine Colonic Hyperplasia was induced in the mice by oral inoculation with a 16-h culture of *C. rodentium* strain DBS100 (10^8^CFUs) identified as pink colonies on MacConkey agar, as previously described [23,70,72]. Age and sex-matched control mice received sterile culture medium only. We employed the standard AIN-93 diets from Envigo (TD.94045) containing, in g/kg: casein 200, DL-methionine 3.0, corn starch 347.311, maltodextrin 130, sucrose 160, soybean oil 70, vitamin mix AIN-93 10, choline bitartrate 2.5, antioxidant TBHQ 0.014, cellulose 50, calcium 12. For the Pectin diet, AIN-93 diet was modified to contain: 335.686 g corn starch, 60 g Pectin (6% Pectin; TD.97202), and no cellulose. Tributyrin diet was procured from TestDiet (6%; Cat#1814961). Mice in the Pectin and Tributyrin group were put on the diet on the third day after CR infection. Mice were then euthanized after 12 days post-infection. For microbiota depletion experiments, mice were given a cocktail of vancomycin (500 mg/L), metronidazole (1 g/L), and ciprofloxacin (0.2 g/L) for 10 days, starting 3 days post-CR infection. Mice were monitored daily by our trained laboratory and veterinary staff to minimize stress and suffering. Animals were immediately euthanized when they showed signs of severe distress, discomfort, loss of weight, and poor prognosis, following consultation with the veterinarians.

### 4.4. Histology and Electron Microscopy

Colon tissues were freshly harvested from mice and fixed with 10% neutral buffered formalin or in Carnoy’s fixative (60% methanol, 30% chloroform, and 10% acetic acid) prior to paraffin embedding. Paraffin-embedded sections (5 μm) were stained with Hematoxylin and Eosin for morphology using standard techniques. Goblet cells were stained with PAS (Richard-Allan Scientific) or Alcian blue (Thermo Fisher Scientific, Waltham, MA, USA) and counterstained with Nuclear Fast Red (Sigma-Aldrich, St. Louis, MO, USA). The pictures were obtained with a Nikon i80 microscope. For electron microscopy, mouse distal colons from uninfected normal mice or *C. rodentium*-infected or *C. rodentium*-infected and diet-treated mice were minced into small cubes, fixed in 4% paraformaldehyde and 2% glutaraldehyde in cacodylate buffer (0.1 M sodium cacodylate, pH 7.6) overnight at room temperature, and postfixed in 1% osmium tetroxide for 90 min. The fixed tissues were dehydrated through a graded series of Ethanols, embedded in Epon-araldite resin, and maintained for 48 h at 60 °C for polymerization. Ultrathin (100 nm) sections cut on a Leica UC-6 ultramicrotome were placed on glow-discharged 300-mesh copper grids and stained with uranyl acetate and Sato’s lead to enhance contrast. Ultrathin sections were examined with a Hitachi H-7600 electron microscope.

### 4.5. Immunohistochemistry

Paraffin-embedded sections were dewaxed with xylene and rehydrated in graded series of ethanol. Tissue sections were then washed with PBS before starting the staining procedure. Immunohistochemistry was done using Histostain-SP kit (Invitrogen, Waltham, MA, USA) following their instructions. Briefly, deparaffinized sections were boiled for 20 min in a 0.01 M citrate buffer at pH 6.0 for epitope retrieval. The sections were then blocked for 10 min using a serum blocking solution. Sections were incubated for 1 h at 4 °C with primary antibodies. After three washings for five minutes each in PBS, sections were subsequently incubated with biotinylated secondary antibodies for 10 min, followed by 10 min incubation with enzyme streptavidin–peroxidase conjugate. A 3–5 min incubation followed this with DAB solution containing diaminobenzidine (DAB) chromagen for development and 0.6% H_2_O_2_ to inhibit endogenous peroxidase activity. The sections were finally counterstained with hematoxylin, dehydrated with a graded series of ethanol, and cleared in xylene. After staining, the sections were mounted with mounting medium, and images were obtained and analyzed with a Nikon 80 microscope.

### 4.6. Immunofluorescence

After dewaxing and rehydration, tissue sections were incubated with relevant primary antibodies overnight at 4 °C. After incubation, the slides were washed three times in PBS for five min each. The secondary antibody staining was performed by covering the tissue sections with a 1:1000 dilution of Alexa Fluor 488/594 conjugated goat anti-rabbit/anti-mouse IgG antibody (original concentration: 2 mg/mL; Thermo Fisher Scientific, USA) in blocking buffer, and the sections were incubated for 1 h at room temperature in the dark. The excess liquid was blotted away, and the sections were rinsed twice for 5 min each using PBS. Next, the sections were stained for 5 min at room temperature in the dark using a 10 mg/mL of DAPI solution diluted in 1× PBS (Sigma-Aldrich, MO, USA). The sections were then rinsed with Milli-Q water and blotted dry. Finally, the sections were covered with ProLong Gold Antifade Mounting media (Invitrogen, Waltham, MA, USA), covered with coverslips, and the edges of the coverslips were sealed with nail polish. The slides were kept at room temperature in the dark for at least 24 h and then visualized by Nikon i80 upright fluorescence microscope.

### 4.7. Fluorescence In Situ Hybridization (FISH)

FISH was performed according to the method described previously with some modification [73]. Paraffin sections were dewaxed and rehydrated in an ethanol gradient to water. The tissue sections were incubated with 5 μg/mL TexasRed-conjugated EUB338 (5′- GCTGCCTCCCGTAGGAGT-3′, Invitrogen) in hybridization buffer (0.1M Tris-HCl, 0.9 M NaCl, 0.1% SDS and 10% formamide, pH 7.2) at 40 °C overnight. The sections were rinsed in washing buffer (20 mM Tris-HCl, 0.9 M NaCl, pH 7.4) at 40 °C for 15 min and stained with 1 μg/mL DAPI. After staining, the sections were mounted with ProLong Gold mounting medium (Invitrogen). All images were obtained and analyzed with a Nikon i80 microscope.

### 4.8. Bacterial DNA Extraction

Fresh feces from mice were used to extract microbial DNA using the QIAmp DNA stool kit (Qiagen, Valencia, CA, USA), following their instructions. The integrity, concentration, and quality of the total DNA were assessed by agarose gel electrophoresis and determined by absorption at A260 and the A260 to A280 ratio, respectively, using a Nanodrop-2000 spectrophotometer (Thermo Fisher Scientific Inc., USA). DNA solutions were stored at −20 °C until further analysis.

### 4.9. Microbial Analysis Using 16S Ribosomal DNA Library Preparation and Sequencing

Using fecal DNA, the hypervariable V3–V4 region of the 16S ribosomal RNA (rRNA) gene was amplified from each sample using barcoded universal bacterial primers and, subsequently, sequencing was done on the Illumina MiSeq platform using custom primers [73]. The resulting sequences were analyzed using the QIIME2 (Quantitative Insights Into Microbial Ecology; http://www.qiime2.org, accessed on 9 November 2021) analysis pipeline [74]. FASTA-quality files and a mapping file indicating that the 8-nucleotide barcode sequence corresponding to each sample was used as inputs. Reads were trimmed and demultiplexed using exact matches to the supplied DNA barcodes. Any reads with homopolymer runs, more than 6 ambiguous bases, nonmatching barcodes, barcode errors, or quality scores less than 25 were removed. Samples with less than 3500 sequences were also removed. The resulting sequences defined at 97% similarity threshold were assigned to operational taxonomic units (OTUs), and taxonomical classification was performed using the greengenes 13_8 database. Core diversity, including alpha and beta diversity analysis, was performed on OTU tables using QIIME software. The resulting raw sequence files (.fastq.gz) were submitted to the NCBI Sequence Read Archive (SRA) database.

### 4.10. Functional Profiling of the Microbial Community

Prediction of functional profiling of microbial communities based on 16S rDNA sequences was performed using Phylogenetic Investigation of Communities by Reconstruction of Unobserved States (PICRUSt) (release 1.0.0) [75]. The OTU tables were normalized by dividing each OTU by the known/predicted 16S rRNA gene copy number abundance. The metagenome functional content was predicted using the Kyoto Encyclopedia of Genes and Genomes (KEGG) Orthology (KO) classification scheme. The predicted metagenome BIOM table was analyzed and visualized using the Statistical Analysis of Taxonomic and Functional Profiles (STAMP) software package v. 2.0.9. The linear discriminant analysis effect size (LEfSe), an algorithm for biomarker discovery that identifies enrichment of abundant taxa or function between two or more groups, was used to compare all taxa at different taxonomic levels simultaneously (i.e., phylum, class, order, family, and genus) between treatment groups.

### 4.11. Fluorescein Isothiocyanate-Dextran (FITC-D) Assay

In vivo permeability assay to assess epithelial barrier function was performed using FITC-D as described [76]. Briefly, food was withdrawn for 4 h from 5-to-6-weeks-old C3H/HeN mice in various groups and gavaged with an 80 mg/100 g body weight of FITC-D, (molecular weight 4000; Sigma–Aldrich). Serum was collected at the time of euthanasia, blood cells were removed by centrifugation, and the fluorescence intensity of each sample was measured with a fluorimeter (excitation, 492 nm; emission, 525 nm; FLUOstar Galaxy 2300; BMG Labtech, Durham, NC, USA). FITC-D concentrations were determined from standard curves generated by the serial dilution of FITC-D, and permeability was calculated by linear regression of sample fluorescence (Excel 5.0; Microsoft).

### 4.12. ChIP-Sequencing and Data Analysis

Young adult mouse colon cells (YAMCs) were cultured and infected as described previously [77]. Cells were infected with CR for 3 h and treated with 5 mM sodium butyrate (Sigma) for 24 h. YAMC with or without CR infection alone were used as a control. The cells were then harvested and subjected to DNA immunoprecipitation using a Magna ChIP A/G kit (Millipore) following the manufacturer’s instructions. Briefly, fresh formaldehyde was added to the media for a final concentration of 1% for 10 min at room temperature to cross-link proteins to DNA. Cross-linking was stopped by the addition of 0.125 mmol/L glycines for five minutes at room temperature. Cells were then lysed in Cell Lysis Buffer containing 1X Protease Inhibitor Cocktail II. Lysates were sonicated on a refrigerated water bath (Bioruptor Standard) for 12 cycles of 20 s on/20 s off at a high setting. The chromatin was sheared to an average length of 200–600 bp and subjected to immunoprecipitation using IgG as a negative control or with specific antibodies against H3K27Ac, H3K4me3, H3K9Ac. ChIP-Sequencing was performed in an Illumina NovaSeq 6000 sequencing machine (Illumina, San Diego, CA, USA) at a 100-base, paired-end read resolution. After read quality assessment and adapter trimming, the sequenced reads were mapped to the mouse genome (mm10) using the Bowtie2 software. Sequencing produced between 26 and 45 million reads per sample, of which 80% to 96% mapped to the reference genome. Significantly enriched ChIP regions (relative to the corresponding input sample) were detected using the model-based analysis of ChIP-Seq (MACS) software.

### 4.13. Lgr5 Promoter Activity Reporter Assay

HEK293 cells were co-transfected in 12 well plates with an Lgr5 promoter–reporter construct and secreted alkaline phosphatase (SEAP) plasmid as normalization control constructs using lipofectamine 2000. After 16 h, fresh media was added along with 1, 5 and 10 mM of sodium butyrate. A total of 50 μL of media was collected after 24 h and luciferase assay was performed using Gaussia Luciferase Assay kit (GeneCopoeia) according to the manufacturer’s instructions.

### 4.14. Cellular Thermal Shift Assay (CETSA) and Western Blotting

Intact, viable SW480 cells that express high levels of Lgr5, grown to 70–80% confluency, were harvested and resuspended to a final concentration of 20 × 10^6^ cells/mL in the plain medium. Cells were divided into different tubes and treated with either water or different concentrations of sodium butyrate followed by incubation at 37 °C for 4 h. A total of 100 L of cell suspension was aliquoted into the PCR plate and heated at temperatures ranging from 43 to 64 °C for 3 min. The plates were then snap-frozen in liquid nitrogen followed by incubation in a water bath three times. All samples were then treated with 4X Laemmeli buffer, boiled, and loaded on SDS-PAGE gel, transferred to a nitrocellulose membrane, and incubated with Lgr5 antibody. Protein levels on Western blot were pictured on ChemiDoc and analyzed by ImageJ software. Densitometry analysis was performed and graphs were drawn.

### 4.15. Molecular Docking

Autodock Vina software program was used for molecular docking [78]. The crystal structure of LGR5 (PDB ID: 4UFR) [79] was downloaded from the protein data bank. Autodock Vina is known for high performance and accuracy for structure-based virtual screening to aid drug discovery. It is freely available software from The Scripps Research Institute (http://vina.scripps.edu/, accessed on 9 November 2021). The 3D-grid box covering all active and crucial residues is generated and assessed by software using established multiple algorithms to predict favorable binding poses. For the present study, a 3D-grid box with a grid center coordinate consisting of grid spacing 1.0 A0 and 60 × 60 × 60 point size was generated, and default parameters of the Autodock tools were used. Both protein and Butyrate were prepared by adding polar hydrogens and total Kollman and Gasteiger charges. Lamarckian GA was used for the prediction of the top 10 conformations. Next, the most stable conformation based of low binding energy, hydrogen bonds, and bond distance were selected and visualized with Pymol (https://pymol.org/2/, accessed on 9 November 2021) [80].

### 4.16. RNA Sequencing

RNA seq was carried out on cultured young adult mouse colon cells (YAMCs) maintained in vitro as previously described [77]. The cells were infected only with CR for 3 h or infected with CR for 3 h then treated with 5mM sodium butyrate (Sigma) for 24 h. Non-infected YAMC cells were used as control. RNA was isolated from the cells using TRIzol (Ambion Life Technologies, Austin, TX, USA) and converted to cDNA using the High-Capacity cDNA Reverse Transcription kit (Applied Biosystems, Waltham, MA, USA). The concentration of RNA was measured using a spectrophotometer (Nanodrop 2000, Thermo Fisher Scientific). About two μg of total RNA for each sample was sent to the genome sequencing facility (University of Kansas Medical Centre, KS) for RNA sequencing. The procedure for RNA sequencing and analysis has been described previously [81]. Briefly, after RNA isolation, subsets of RNA become isolated for polyA transcripts, whereupon they are then converted to cDNA. This is followed by the addition of sequencing adaptors to the ends of the cDNA fragments before amplification by PCR to generate the RNA-sequencing library, which is proceeded by downstream transcriptome analysis. YAMC–control, YAMC–*Citrobacter rodentium*, and YAMC–*Citrobacter rodentium* + sodium butyrate samples were sequenced in biological triplicates. Sequencing was generated between 25.1 and 28.6 million reads per sample. The read quality was assessed using the FastQC software. The reads were then mapped to the mouse genome (GRCm38. rel98) using the STAR software, version 2.3.1z. Between 97% and 98% of the sequenced reads were mapped to the reference genomes in the 9 samples, resulting in between 24.6 and 28.0 million mapped reads per sample, of which, on average, 92.5% were uniquely mapped reads. Transcript abundance estimates were calculated using the featureCounts software. Expression normalization and differential gene expression calculations were performed in DESeq2 software to identify statistically significant, differentially expressed genes. 

### 4.17. Statistical Analysis

The values are expressed as mean ± SD. Statistical analyses of all studies were performed using unpaired, two-sided Student’s *t*-tests or one-way analysis of variance (ANOVA) models for multiple-group comparisons. *p* < 0.05 was considered statistically significant. The sample size was determined based on the results from pilot studies with similar mouse numbers. All experiments used *n* = 5 mice per group unless otherwise indicated and represent 2–3 independent experiments. For in vivo studies, the reported sample number per experiment represents biologic replicates (comparing animals of one genotype with littermates of another genotype). Both male and female mice were used in the study. All analyses were performed using GraphPad, version 9.

## 5. Conclusions

In conclusion, this study delineates the ability of Pectin and Tributyrin diets to alleviate colitis through microbiome and mucin restoration, Lgr5+ cell expansion, and reduced expression of proinflammatory markers. This work highlights the crosstalk between diet, microbiota, and the colon stem cells working together to help restore the crypt structure following an insult.

## Figures and Tables

**Figure 1 ijms-23-00339-f001:**
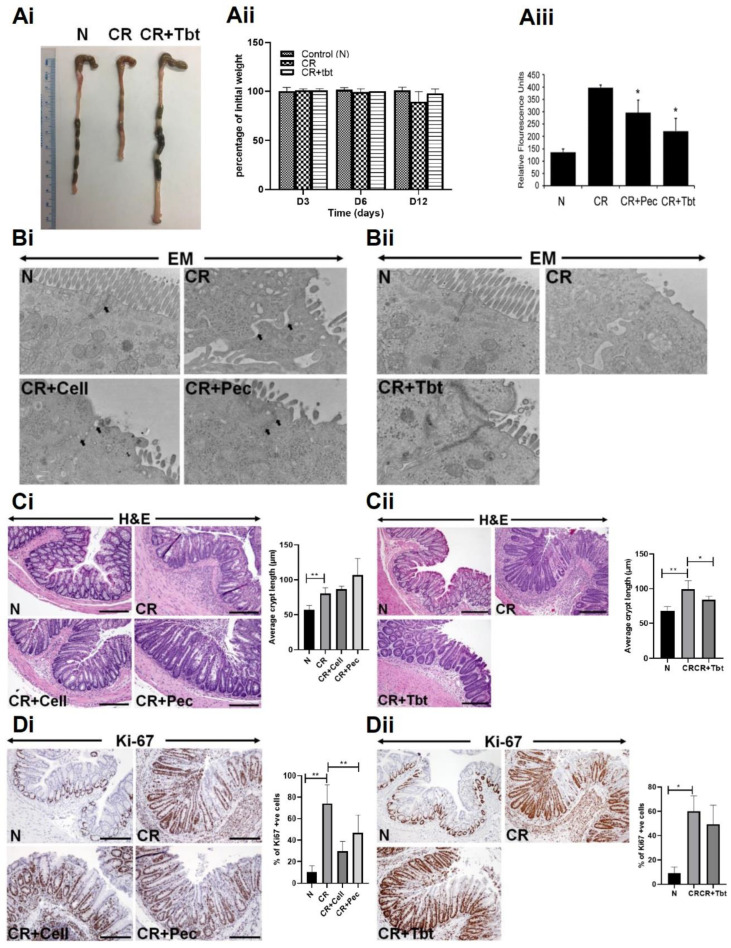
Pectin and Tributyrin diets ameliorate infectious colitis: (**Ai**). Representative images of colons from uninfected normal mice (N), *C. rodentium*-infected (CR) and CR-infected, and Tributyrin diet-treated mice (CR+Tbt) (results represent 3 independent experiments). (**Aii**). Average weight measurement of mice in various treatment groups from D0 to D12 post-infection (*n* = 5 per group). (**Aiii**). Normal, CR, CR, and Pectin-treated (CR+Pec), and CR and Tributyrin-treated (CR+Tbt) mice were subjected to gavage with FITC-D, and their serum concentrations were measured after four hours (results represent 3 independent experiments; *p* < 0.05 compared to CR; Mean+SD) (**Bi**,**Bii**). Electron microscopy (EM) of the mice is described in Panel A. Arrows represent tight junctions (Pec, Pectin; Cell, Cellulose; Tbt, Tributyrin). (**Ci**,**Cii**). H&E staining in the colon sections prepared from the indicated groups in C3H mice. Arrows indicate immune cell infiltration in the CR group. Bar graphs represent average crypt length (* *p* < 0.05; ** *p* < 0.01 compared to N; Mean+SD; two-tailed Student’s *t*-test). (**Di**,**Dii**). Representative staining for Ki-67 in the colon sections prepared from the indicated groups in C3H mice. Scale bar = 100 μm; Bar graphs represent percent of Ki-67 positive cells; (* *p* < 0.05; ** *p* ≤ 0.01); two-tailed Student’s *t*-test; results represent 3 independent experiments.

**Figure 2 ijms-23-00339-f002:**
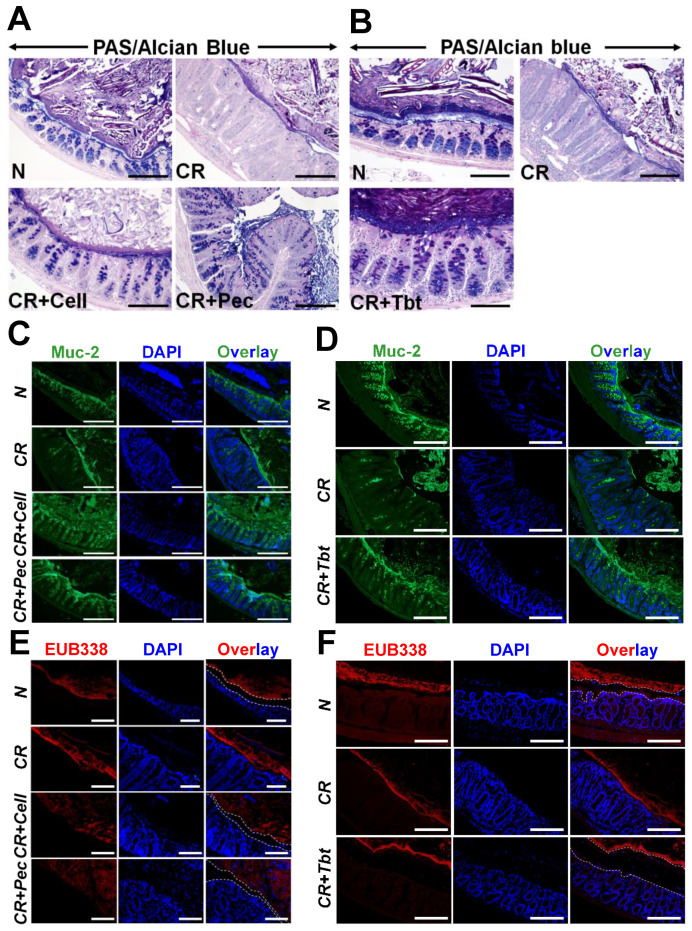
Pectin and Tributyrin diets restore mucus layers and restrict bacterial invasion: (**A**,**B**). PAS/Alcian blue staining in the colon sections prepared from the indicated groups in C3H mice. (**C**), (**D**). Representative immuno-staining for Muc2 in the colon sections prepared from the indicated groups in C3H mice. Bar = 100 μm; *n* = 5 mice/group. (**E**). Representative EUB338 staining for bacteria in the Carnoy-fixed colonic tissues of N, CR, CR+Cell, CR+Pec mice (C3H) via FISH (TexasRed-EUB338; Scale bar = 75 μm; *n* = 5 mice/group). DAPI was used as a counterstain. (**F**). Representative EUB338 staining for bacteria in the Carnoy-fixed colonic tissues of N, CR, CR+Tbt mice (C3H) via FISH (TexasRed-EUB338; Scale bar = 100 μm; *n* = 5 mice/group). DAPI was used as a counterstain. Please note how both Pectin and Tributyrin diets restrict bacterial movement near the mucosa, marked by dashed lines. N= normal mice/control; *C. rodentium*-infected (CR); mice infected with CR and fed a cellulose diet (CR+Cell); mice infected with CR and fed a Pectin diet (CR+Pectin); mice infected with CR and fed with Tributyrin diet (CR+Tributyrin).

**Figure 3 ijms-23-00339-f003:**
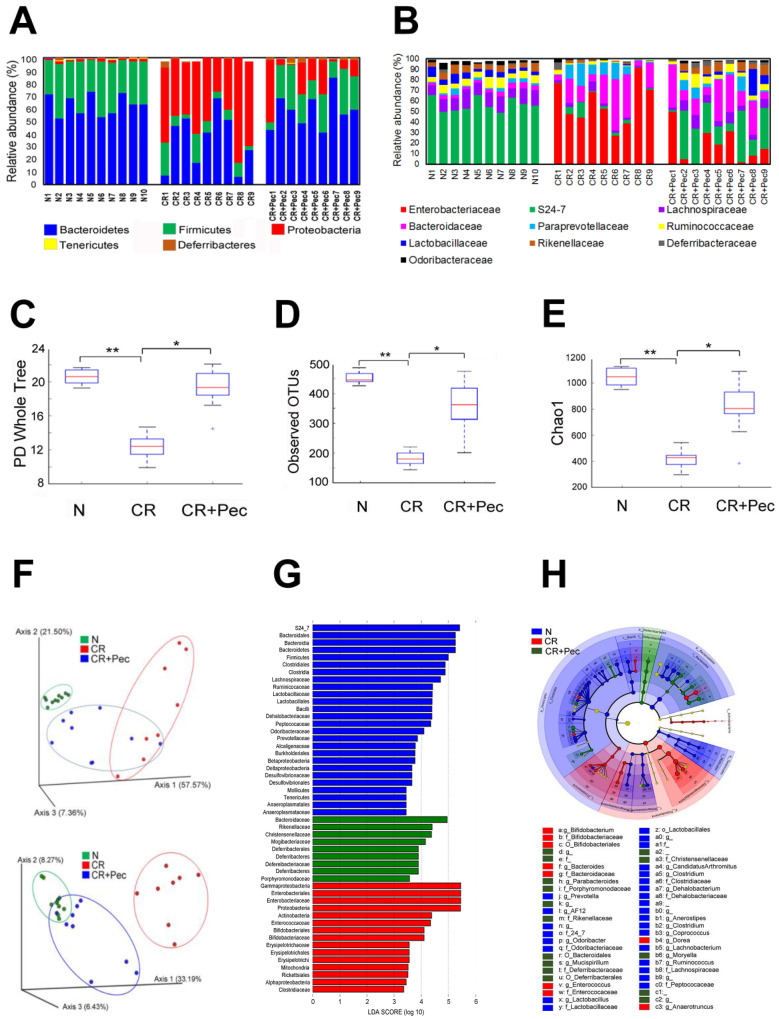
Dietary intervention ameliorates infection-induced microbial dysbiosis: (**A**). The relative abundance of OTUs at the phylum level in fecal samples of control (N), CR (*C. rodentium*-infected), and mice infected with CR and fed a Pectin diet (CR+Pectin). (**B**). The relative abundance of OTUs at the family level in control, CR, and CR+Pectin fecal samples. The data are based on 16S rRNA and show the most abundant OTUs in 9 samples. (**C**–**E**). Box plots of N, CR, and CR+Pectin fecal samples showing Alpha diversity indices (Chao1, observed OTUs, and PD whole tree; * *p* < 0.05, ** *p* < 0.01; one-way ANOVA followed by Tukey’s test). **F**. Principal component analysis (PCoA) based on the bacterial community structure using weighted and unweighted UniFrac distance in fecal samples from N, CR, and CR+Pectin mice. (**G**). Linear discriminant analysis (LDA) identifies the most differentially abundant taxa among N, CR, and CR+Pectin samples. (**H**). Linear discriminant analysis Effect Size (LEfSe) cladogram demonstrating taxonomic differences in N, CR, and CR+Pectin microbiota.

**Figure 4 ijms-23-00339-f004:**
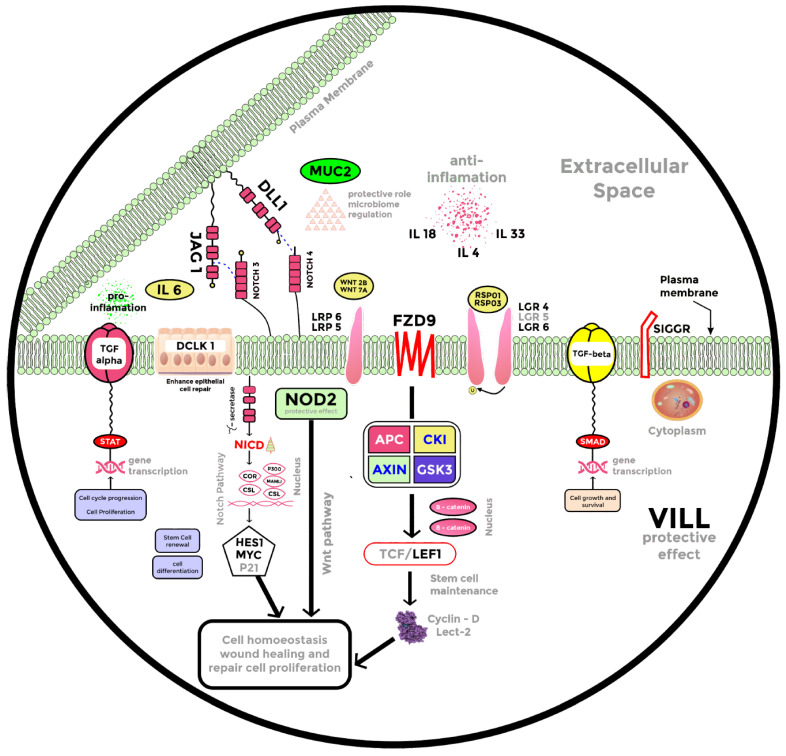
RNA-seq in young adult mouse colon (YAMC) cells.

**Figure 5 ijms-23-00339-f005:**
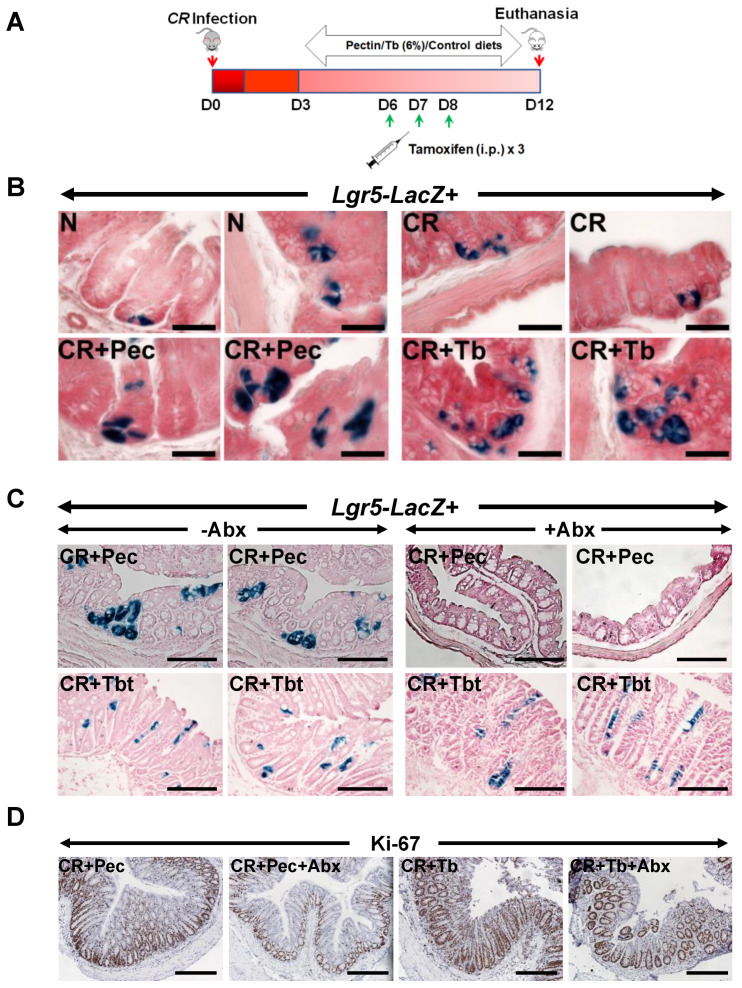
Requirement of microbiota for Lgr5 expression: (**A**). Experimental approach to perform lineage tracing in *Lgr5-EGFP-IRES-CreERT2*/*Rosa26LacZ* reporter (*Lgr5-R*) mice following CR infection and dietary and/or tamoxifen treatment. (**B**). Representative images of *Xgal* stained crypt sections from N, CR, CR+Pec, and CR+Tbt-treated *Lgr5-R* mice (Scale bar = 75 μm; results represent 3 independent experiments). (**C**). Representative images of *X*gal-stained crypt sections from either CR+Pec and CR+Pec+Abx or CR+Tbt and CR+Tbt+Abx-treated *Lgr5-R* mice (Scale bar = 150 μm; results represent 3 independent experiments). (**D**). Representative staining for Ki-67 in the colon sections of *Lgr5-R* mice treated with either Pectin and Pectin+Abx or Tributyrin and Tributyrin+Abx (results represent 3 independent experiments). N = normal mice/control; *C. rodentium*-infected (CR); mice infected with CR and fed a Cellulose diet (CR+Cell); mice infected with CR and fed a Pectin diet (CR+Pectin); mice infected with CR and fed with Tributyrin diet (CR+Tributyrin); Abx (antibiotics); Lgr5-Lacz+ (mice with positive staining for *LacZ*+ve progeny of the Lgr5+ve stem cells were visualized in the colon by β-galactosidase staining). Scale bar = 120 μm; results represent 3 independent experiments.

**Figure 6 ijms-23-00339-f006:**
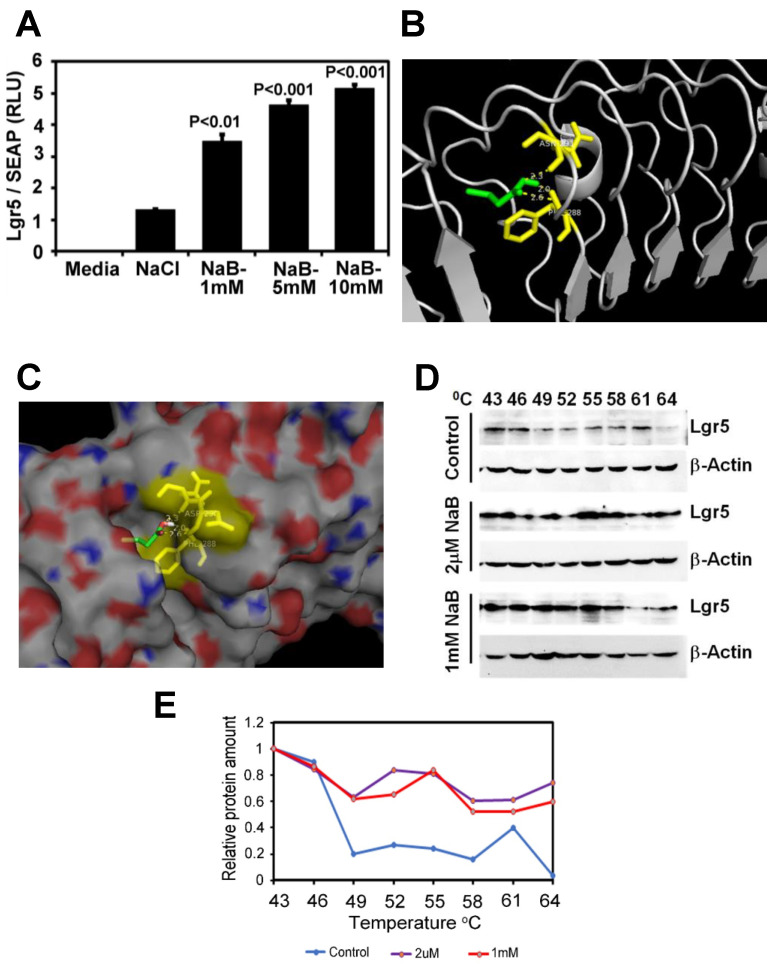
Butyrate directly binds and regulates thermal stability of Lgr5: (**A**). Lgr5 promoter reporter activity assay in HEK293 cells treated with different concentrations of sodium butyrate (P values compared to NaCl; results represent 3 independent experiments). (**B**,**C**). Ribbon (**B**) or space-filled (**C**) models of molecular docking revealed that Butyrate binds within the protein cavity of Lgr5 (binding energy = −4.0 Kcal/mol) and forms hydrogen bonds with Asp290 (2.3 Å) and Phe288 (2.0 and 2.6 Å). Green: Butyrate; Yellow: Interacting amino acids; Red–White–Blue: Lgr5 protein surface view. (**D**). Western blot of HEK293 cells treated with different concentrations of sodium butyrate for two hours and then subjected to thermal denaturation at different temperatures for 3 min. Cell lysates were prepared and following SDS-PAGE membranes were probed with Lgr5 antibody. Please note Lgr5 stabilization at higher temperatures when incubated with different concentrations of Butyrate compared to control. (**E**). Results from the densitometric evaluation of CETSA assays. CETSA: Cellular Thermal Shift Assay, Secreted Alkaline Phosphatase (SEAP); NaB: Sodium Butyrate; NaCl: Sodium Chloride.

**Figure 7 ijms-23-00339-f007:**
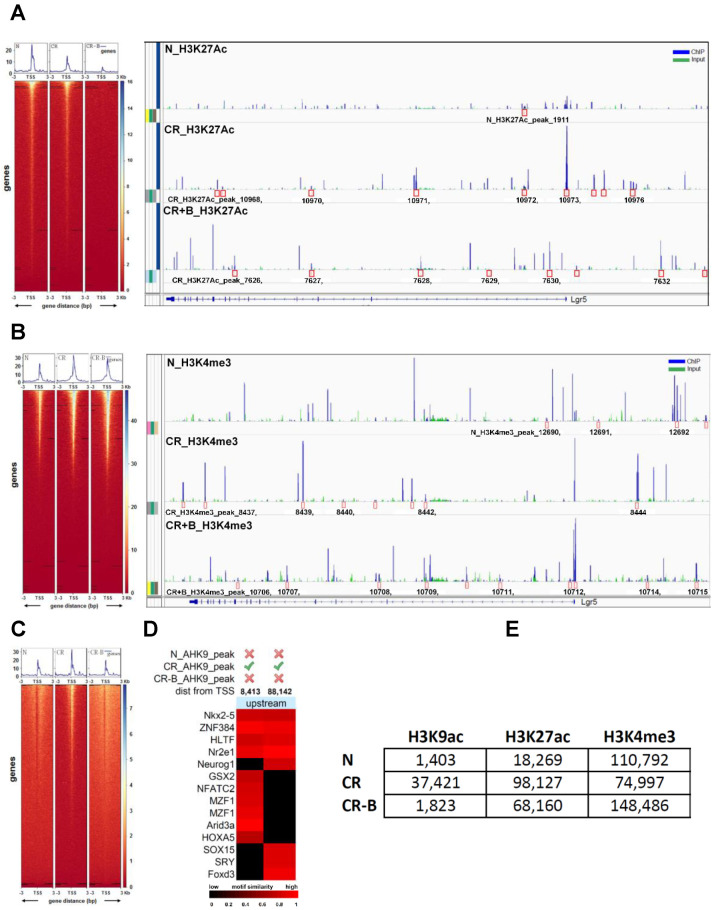
Chromatin immunoprecipitation and sequencing (ChIP-seq) analysis in YAMC cells showing the number of significantly enriched regions detected for each histone modification marker: (**A**). Heatmap of H3K27Ac sequencing-depth-normalized read coverage around 3 kb upstream and downstream from the transcription start site (TSS) of Lgr5 (top panel); sites enriched for the H3K27Ac marker around a 100 kb region from the TSS of the Lgr5 gene in N, CR, and CR+Butyrate (bottom panel). (**B**). Heatmap of H3K4me3 sequencing-depth-normalized read coverage around 3 kb upstream and downstream from the transcription start site (TSS) of Lgr5 (top panel); sites enriched for the H3K4me3 marker around a 100 kb region from the TSS of the Lgr5 gene in N, CR, and CR+Butyrate (bottom panel); (**C**). Heatmap of H3K9Ac sequencing-depth-normalized read coverage around 3 kb upstream and downstream from the transcription start site (TSS) of Lgr5. (**D**). Two upstream sites (8413 bp and 88,142 bp) enriched for H3K9ac in CR that contained conserved binding motifs for several known TFs. (**E**). The number of significantly enriched regions detected for each histone modification marker in several groups.

**Table 1 ijms-23-00339-t001:** List of designated genes differentially regulated in response to either CR infection or CR infection coupled with Butyrate treatment shown here as fold change (representative of 3 biological replicates).

Genes	Fold Change CR Infection vs. Control	Fold Change Butyrate Treatment vs. Control	Fold Change Butyrate Treatment vs. CR Infection
*DCLK 1*	−2.5	−1.1	2.3
*DLL1*	−1.8	1.5	2.7
*FZD9*	2.2	6.1	2.8
*HES 1*	−1.2	1.2	1.4
*IL 18*	−2.1	−1.3	1.6
*IL 33*	−1.7	1.8	3.1
*IL 4*	1.1	2.3	2.1
*IL 6*	3.2	2.1	−1.5
*JAG 1*	−1.1	4.6	5.1
*LEF 1*	1.4	10.4	7.5
*LGR 4*	−1	2.2	2.2
*LGR 6*	1.6	9.1	5.7
*LRP 5*	−1	1.5	1.5
*LRP 6*	−1.1	1.5	1.7
*MUC 2*	−1	8.7	9.1
*MYC*	1.7	2.3	1.4
*NOD 2*	1.5	12.2	7.9
*NOTCH 3*	−1.2	10.6	12.3
*NOTCH 4*	−1.6	4.5	7.4
*RSP0 1*	−3.4	−1.3	2.6
*RSP0 3*	−1.2	2.8	3.4
*SIGIRR*	1.8	28.9	16.3
*TGF-Alpha*	2.1	21.1	10.2
*TGF-Beta*	−1.7	10	17
*VILL*	−1.9	10.6	20.5
*WNT 2B*	−3.1	−1.3	2.3
*WNT 7A*	1.4	52.6	36.3

Schematic of the proposed physiological changes with Butyrate treatment after CR infection, this model was used to suggest a possible role of SCFAs such as Butyrate in mitigating CR-induced colitis. The image includes selected cytokines, genes involved in stem cell maintenance, protective effects, cellular homeostasis, and wound healing.

## Data Availability

All the data are contained within the manuscript. For the 16S gene sequencing, the raw sequence files (.fastq.gz) were submitted to the NCBI Sequence Read Archive (SRA) database and can be accessed here: https://www.ncbi.nlm.nih.gov/sra/PRJNA757697, accessed on 9 November 2021.

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
