# Peer review of "Dietary Interventions Ameliorate Infectious Colitis by Restoring the Microbiome and Promoting Stem Cell Proliferation in Mice"

_ijms, 2021, doi:10.3390/ijms23010339_

Round 1
Reviewer 1 Report
The authors present a very interesting study. The decrease in short-chain-fatty-acids (SCFAs) is linked to inflammatory bowel disease (IBD). However, the mechanisms through which SCFAs promote wound healing, orchestrated by intestinal stem cells, are poorly understood. The authors discovered that in mice with Citrobacter rodentium (CR) induced infectious colitis, treatment with Pectin and Tributyrin diets reduced the severity of colitis by restoring Firmicutes and Bacteroidetes and by increasing mucus production. RNA-seq in Young Adult Mouse Colon (YAMC) cells identified higher expression of Lgr4, Lgr6, DCLK1, Muc2, and 18 SIGGIR after butyrate treatment. A multitude of effects on the gut microbiome, differential gene expression and/or expansion of Lgr5(+) stem cells, seem to underlie amelioration of colitis following the dietary intervention. With regard to the CR model. Citrobacter rodentium (CR) is a Gram-negative enteric bacterium that causes Transmissible Murine Colonic Hyperplasia (TMCH) in mice characterized by colon hyperplasia, loss of goblet cells, and increased inflammation depending upon the genetic background. The authors should expand in the discussion the role of NLRP3 in CR (https://pubmed.ncbi.nlm.nih.gov/32841451/).
In numerous figures, it seems that the scale bar has not been labeled properly. I would strongly suggest the authors control the original figures and which magnification was used for the microphotographs.
Reviewer 2 Report
This manuscript provides informative microbiome and transcriptome profiles from prebiotic diet-treated mice and cell line with beautiful representative pictures. However, there are several major scientific concerns.
Major comments
- Most figures lack statistical analyses: The author shows only representative pictures in the main figures. ie. Figure 1Aii, C, D, Figure 2, Figure 3 A, B, Figure 4 top panel, Figure B-D, Figure 6E, Figure S1. Should show bar graph with statistic analyses as well. The author can show positive cells/crypt etc.
- Most figure legends lack explanation of statistics: *means p<0.05?, **means p<0.01?, what type of method (ie. t-test or Tukey with ANOVA)?
- Most figure legends lack explanation of abbreviations: for example, in Figure2 CR+Cell means cellurose?; in Figure 3, CR+Pectin means CR+Pec in the figure? N means non-treatment? ; in Figure 5, Abx means antibiotics? Lgr5-LacZ+ means Lgr5-R with LacZ staning?; in Figure 6, SEAP? NaB?
- All supplementary figures lack legends.
- Methods section lacks several methodologies: Figure 1 Bi Bii: I could not find the method for “Electron microscopy” in Methods section. Please provide the details for “Electron microscopy” or indicate it clearly in Methods section.
- Methods section lacks several methodologies: I could not fine the information for Pectin and Tributyrin. Please provide the details: ie. concentration that you used, vender info, cat#, how to add these to normal diet etc.
- Supplementary Fig1: The author used C3H for all experiments. Why did you use C57 Bl/6 mice for Fig S1? Please clarify the issue.
- Is the order of Method section correct? Please check the instruction of the journal.
- Also check the order of reference. The order of reference is not correct. Looks like the author moved Methods section to the back of Results section.
- Method 4.17: I could not find PCR results. Please clarify that. If you show this method for PCR, please indicate “method for quality check” like DNA quality check you showed before.
Minor comments
- Line 30: scfas => SCFAs
- Figure 1 legend: The sentence “n=3 independent experiments” is not clear. Please indicates mice number you used and/or describe “the results represent three independent experiments” etc.
- Figure 1 legend: 100 mm ==> 100µm?
- Figure 1 Aii: The difference of start body weight looks strange. Please show % of initial.
- Method, animals: When did you start special diet? after colonization of CR or same time at the colonization? It may affect colonization of CR. Please indicate details.
- Method, animals: The author uses C57 Bl/6 mice for Fig S1. Please provide the info.
- Figure 2: use same font size for C-F.
- Figure 3 legend: *p<0.05? **p<0.01?
- Line 188: Spell out LDA
- Figure 4 top panel: How many replications did you used? Is the number average or representative? Please indicate that in the legend or method.
- Figure 4 bottom panel: Remove red lines (like spell-check)
- Figure 5-7: Remove dots. A. => A
- Figure 5 A: Is CR Infection D0 or D1? Please indicate red arrow correctly.
- Figure 6 A: Is p<0.01 compared to what? Please explain in the legend.
- Figure 6 B and C: What’s your point in figure 6 B and C? Please explain more in the legend or text for persons who are not familiar with the figures.
- Figure 6D: I could not find the details of method. What’s the target of the band? Complex of lrg+butyrate? Please explain the details in the Methods section.
- Line 302: “We performed ChIP-sequencing....” in YAMC study? Please indicate the exact protocol.
Author Response
Comments and Suggestions for Authors
This manuscript provides informative microbiome and transcriptome profiles from prebiotic diet-treated mice and cell line with beautiful representative pictures. However, there are several major scientific concerns.
Major comments
Most figures lack statistical analyses: The author shows only representative pictures in the main figures. ie. Figure 1Aii, C, D, Figure 2, Figure 3 A, B, Figure 4 top panel, Figure B-D, Figure 6E, Figure S1. Should show bar graph with statistic analyses as well. The author can show positive cells/crypt etc.
In the revised manuscript, we have revamped the statistical analysis and calculated either crypt length or number/% of cells positive for each marker. Wherever needed, we have also provided bar graphs as requested. For Figure 4A, these are relative abundances and therefore we presented them as actual numbers. For Figure 6E, since the samples are not in replicates, we feel that an unpaired t-test is less likely.
Most figure legends lack explanation of statistics: *means p<0.05?, **means p<0.01?, what type of method (ie. t-test or Tukey with ANOVA)?
We apologize for this mistake. We have now explained statistical analysis for each figure in the revised manuscript as requested by the reviewer.
Most figure legends lack explanation of abbreviations: for example, in Figure 2 CR+Cell means cellurose?; in Figure 3, CR+Pectin means CR+Pec in the figure? N means non-treatment? ; in Figure 5, Abx means antibiotics? Lgr5-LacZ+ means Lgr5-R with LacZ staning?; in Figure 6, SEAP? NaB?
As requested, all the abbreviations have been explained in the revised manuscript.
All supplementary figures lack legends.
We now provide the figure legends for all supplementary figures.
Methods section lacks several methodologies: Figure 1 Bi Bii: I could not find the method for “Electron microscopy” in Methods section. Please provide the details for “Electron microscopy” or indicate it clearly in Methods section.
We have now provided details of the Electron Microscopy methodology in the revised manuscript as requested by the reviewer.
Methods section lacks several methodologies: I could not find the information for Pectin and Tributyrin. Please provide the details: ie. concentration that you used, vender info, cat#, how to add these to normal diet etc.
All the details regarding Pectin and Tributyrin diets including concentration used, vender info, cat#, etc., have been provided in the revised manuscript.
Supplementary Fig1: The author used C3H for all experiments. Why did you use C57 Bl/6 mice for Fig S1? Please clarify the issue.
We agree with the reviewer that we should have performed 16S rRNA gene sequencing in C3H mice as well. We acknowledge this as a limitation of the study but due to time constraints, will continue the study as a follow-up.
Is the order of Method section correct? Please check the instruction of the journal. Also check the order of reference. The order of reference is not correct. Looks like the author moved Methods section to the back of Results section.
We have checked the journal format and confirm that the manuscript section should be arranged in this order: Introduction, Results, Discussion, Materials and Methods, Conclusions (Optional).
Method 4.17: I could not find PCR results. Please clarify that. If you show this method for PCR, please indicate “method for quality check” like DNA quality check you showed before.
We apologize for this error. We have removed the section on PCR.
Minor comments
- Line 30: scfas => SCFAs
- Figure 1 legend: The sentence “n=3 independent experiments” is not clear. Please indicates mice number you used and/or describe “the results represent three independent experiments” etc. ==> This has been corrected in the revised manuscript.
- Figure 1 legend: 100 mm ==> 100µm
- Figure 1 Aii: The difference of start body weight looks strange. Please show % of initial ==> This has been corrected in the revised manuscript.
- Method, animals: When did you start special diet? after colonization of CR or same time at the colonization? It may affect colonization of CR. Please indicate details. ==> We thank the reviewer and agree for this comment. All interventions in this model began on day 3 post-infection.
- Method, animals: The author uses C57 Bl/6 mice for Fig S1. Please provide the info. ==> This information has been updated in the Methods section.
- Figure 2: use same font size for C-F. ==> We now provide figures with the same font sizes.
- Figure 3 legend: *p<0.05? **p<0.01? ==> Results are expressed as the mean ± SEM and statistical analysis was performed with one-way ANOVA followed by Tukey's test.
- Line 188: Spell out LDA ==> This has been modified as suggested.
- Figure 4 top panel: How many replications did you used? Is the number average or representative? Please indicate that in the legend or method. ==> This has been corrected as requested.
- Figure 4 bottom panel: Remove red lines (like spell-check) ==> This has been corrected as requested.
- Figure 5-7: Remove dots. A. => A ==> all the dots have been removed as suggested.
- Figure 5 A: Is CR Infection D0 or D1? Please indicate red arrow correctly. ==> We have modified Figure 5A to show that CR infection began at D0.
- Figure 6 A: Is p<0.01 compared to what? Please explain in the legend. The p value in the data sets was obtained from two tailed unpaired t-test comparing the treatment groups with the NaCl control.
- Figure 6 B and C: What’s your point in figure 6 B and C? Please explain more in the legend or text for persons who are not familiar with the figures. ==> This has been provided in the revised manuscript.
- Figure 6D: I could not find the details of method. What’s the target of the band? Complex of Lgr5+butyrate? Please explain the details in the Methods section. ==> We now provide details of Cellular thermal shift assay (CETSA) to study protein stability at varying temperatures. This section has been modified.
- Line 302: “We performed ChIP-sequencing....” in YAMC study? Please indicate the exact protocol. ChIP-sequencing refers to Chromatin Immunoprecipitation followed by sequencing of the DNA material. This detail is provided in the Methods section under 4.12.
Reviewer 3 Report
The authors have evaluated the efficacy and modes of action of Pectin or Tributyrin in the treatment of Citrobacter rodentium (CR) colitis in mice and of butyrate on gene expression in CR-treated Young Adult Mouse Colon (YAMC) cells. Dietary treatment with Pectin or Tributyrin diets reduced the severity of CR-colitis by restoring Firmicutes and Bacteroidetes and by increasing mucus production. Lgr5+ stem cell expansion was enhanced in CR mice given Pectin or Tributyrin compared to levels in mice given control diet. Lgr5+ expansion was greatly reduced in Pectin-fed GM mice treated with antibiotics but not in Tributyrin-fed mice.
Butyrate treatment of CR-infected YAMC cells increased Lgr4, Lgr6, DCLK1, Muc2, and SIGGIR expression, and elevated Lgr5 promoter reporter activity possibly due to modulation of a transcriptional activator of Lgr5 (SPIB).
This well-conducted and thorough study clearly demonstrate that effects on the gut microbiome, differential gene expression and/or expansion of Lgr5(+) stem cells, seem to underlie the ameliorative effects of Pectin on CR-colitis.
The basic assumption is that the protective effects of Pectin are due to butyrate production but the differences in responses to Pectin or Tributyrin in vivo and between Pectin in vivo and butyrate in vitro on Lrg5+ expression suggest a more complex process. Do other factors or metabolites add to the actions of butyrate? Does Pectin induce rapid changes in microbiota that are not evident with Tributyrin and thereby aid recovery? LRG5+ expression is driven by many factors produced in the myofibroblast sheath that underlies the epithelium. It provides signals that regulate cell proliferation, cell movement and lineage commitment and maturation. Is this layer disrupted by CR and protected and restored by dietary Pectin? The authors should discuss these options.
Ln 100 Why is there no picture of the colon from CR+Pec or body weight data for that test group?
Why was a cellulose group included in this analysis? Need to specify what Cell group means at this point as it does not become clear until reaching the materials and methods.
Ln 146-151 Microbiome of faeces was analysed. Any measurements on bound and unbound bacteria in the colon? Colon-bound bacteria are sometimes considered to be a better marker of likely IBD severity.
Were faecal SCFAs determined?
Ln 152 Which regions of the colon (ascending, transverse, descending) were sampled or was a complete representative strip of the whole colon used.
Ln 198-199 Could be argued that Pectin prevented loss of key bacteria rather than restored diversity or in fact does both.
Ln 251 See general comments on Lrg5 and Pectin or butyrate.
Ln 266 What were the effects of antibiotics on the microbiome? Did detectable numbers persist in the gut? What effect did antibiotics alone have on host responses to CR?
Ln 289 Fig 6D What housekeeper protein was used?
Ln 336 ‘stem cell markers Lgr4, Lgr5, and Lgr6’. Lgr4-, Lgr5-, and Lgr6-positive stem cells?
Ln 369 ‘Pectin diet boosted the growth of healthy’. Pectin diet boosted the growth and/or retention of healthy?
Also, can the protective actions of Pectin definitely be ascribed to butyrate alone?
Ln 444-447 Is this suggested property of SCFAs necessarily a good thing?
Ln 454 Can protective effects be exclusively linked to butyrate.
Ln 469 Cellulose group. Why was this group included? Need some explanation on Cell group at Ln 100 Fig 1. It appears in the figure but with no indication of what Cell refers to.
Ln 491-494 Were antibiotic studies 14 days in length, as opposed to 12 days for others.
Round 2
Reviewer 2 Report
Thank you very much for revising the manuscript. It’s been improved very much. I have several minor comments as below..
- It was slightly hard to review this revised version because several corrections were NOT highlighted. In addition, old figures and sentences were still there in the revised version. If possible, please submit a simple version without revision history, but with your changes highlighted.
- Figure 1 legend: “Ai”(bold) ==> not bold
- Figure 1 Ai: CR+Tb ==> CR+Tbt
- Figure 1 Aiii: * means p<0.05? Please explain that in the legend.
- Figure 1 Aiii: * is compared to what? Please explain that in the legend.
- Figure 1 Bi-Bii: Spell out EM in the legend.
- Figure 1 Bi-Bii: CR+Cell ==> spell out in the legend: Cell=cellulose.
- Figure 1 legend: p≦ ==> p <
- Figure 1 legend for Ci-Dii bar graph: Please clarify that * and ** mean what? *p<0.05? **p<0.01?
- Figure 1 legend for Ci-Dii bar graph: Please clarify that bar graphs show mean or median? and SD or SEM?
- From Line 128: Are reference numbers correct? [39,40] should be [27,28]. Please check the reference numbers again.
- Line 578: C. rodentium ==> rodentium (Italic style)
- I could not see the legend for supplemental figure that you added. Please clarify that.
Author Response
Response to the Reviewer’s Comments
We thank the reviewer for a careful assessment of our manuscript. We have tried to respond as accurately as possible and hope that the revised manuscript will be deemed acceptable.
- It was slightly hard to review this revised version because several corrections were NOT highlighted. In addition, old figures and sentences were still there in the revised version. If possible, please submit a simple version without revision history, but with your changes highlighted.
We apologize for the lack of insight when submitting the revised manuscript. We have now highlighted in yellow all the corrections and removed the old figures. The manuscript appears more legible now.
- Figure 1 legend: “Ai”(bold) ==> not bold
Corrected as requested
- Figure 1 Ai: CR+Tb ==> CR+Tbt
Corrected as requested
- Figure 1 Aiii: * means p<0.05? Please explain that in the legend.
Explained as requested
- Figure 1 Aiii: * is compared to what? Please explain that in the legend.
Explained as requested
- Figure 1 Bi-Bii: Spell out EM in the legend.
Corrected as requested
- Figure 1 Bi-Bii: CR+Cell ==> spell out in the legend: Cell=cellulose.
Corrected as requested
- Figure 1 legend: p≦ ==> p <
Corrected as requested
- Figure 1 legend for Ci-Dii bar graph: Please clarify that * and ** mean what? *p<0.05? **p<0.01?
Clarified in the revised manuscript
- Figure 1 legend for Ci-Dii bar graph: Please clarify that bar graphs show mean or median? and SD or SEM?
Clarified in the revised manuscript
- From Line 128: Are reference numbers correct? [39,40] should be [27,28]. Please check the reference numbers again.
We thank the reviewer for pointing this out. We have corrected the references in the revised manuscript.
- Line 578: C. rodentium ==> rodentium (Italic style)
Corrected as requested
- I could not see the legend for supplemental figure that you added. Please clarify that.
Please note that the legends for Supplementary Figures were provided in the original manuscript and they can be seen above the Figures. We have now highlighted them in yellow so that they are more visible.